# Influenza-virus membrane fusion by cooperative fold-back of stochastically induced hemagglutinin intermediates

**Tijana Ivanovic[1,2]\*, Jason L Choi[1], Sean P Whelan[3], Antoine M van Oijen[1†], Stephen C Harrison[1,4]\***

[1]Department of Biological Chemistry and Molecular Pharmacology, Harvard Medical School, Boston, United States; [2]Department of Molecular, Cellular and Developmental Biology, University of Colorado, Boulder, Colorado, United States; [3]Department of Microbiology and Immunology, Harvard Medical School, Boston, United States; [4]Howard Hughes Medical Institute, Harvard Medical School, Boston, United States

**\*For correspondence:** ivanovic@crystal.harvard.edu (TI); harrison@crystal.harvard.edu (SCH)

[†]**Present address:** Zernike Institute for Advanced Materials, University of Groningen, Groningen, The Netherlands

**Competing interests:** The authors declare that no competing interests exist

**Abstract** Influenza virus penetrates cells by fusion of viral and endosomal membranes catalyzed by the viral hemagglutinin (HA). Structures of the initial and final states of the HA trimer define the fusion endpoints, but do not specify intermediates. We have characterized these transitions by analyzing low-pH-induced fusion kinetics of individual virions and validated the analysis by computer simulation. We detect initial engagement with the target membrane of fusion peptides from independently triggered HAs within the larger virus-target contact patch; fusion then requires engagement of three or four *neighboring* HA trimers. Effects of mutations in HA indicate that withdrawal of the fusion peptide from a pocket in the pre-fusion trimer is rate-limiting for both events, but the requirement for cooperative action of several HAs to bring the fusing membranes together leads to a long-lived intermediate state for single, extended HA trimers. This intermediate is thus a fundamental aspect of the fusion mechanism.

## Introduction

Fusion of two lipid-bilayer membranes is a thermodynamically favorable process, but it crosses a high kinetic barrier as the two bilayers approach each other. Efficient fusion therefore requires a catalyst, a role served in living cells by a fusion protein or protein complex. The influenza virus hemagglutinin (HA) has become an important paradigm of a fusion catalyst, in part because of early crystallographic and mechanistic studies and in part because of continued concern about a virus that caused tens of millions of deaths during the 20th century. HA facilitates fusion by undergoing a large-scale conformational change, coupled to the two fusing membranes (virus and target).

Our current picture of HA-mediated membrane fusion, illustrated in *Figure 1A* (*Harrison, 2008*), comes from HA structures in both pre- and post-fusion conformations and from inferences about transient intermediate states. HA is a homotrimer, synthesized as an inactive precursor, $[HA_0]_3$, and activated for fusion by proteolytic cleavage of each chain into $HA_1$ and $HA_2$, yielding $[HA_1\text{-}HA_2]_3$. At the N-terminus of $HA_2$ is a hydrophobic 'fusion peptide', which following cleavage inserts firmly into a pocket near the axis of the trimer (*Chen et al., 1998*). Exposure to low pH, which during infection occurs in an endosome, causes the $HA_1$ 'head' to separate from the $HA_2$ 'stem' and enables a set of $HA_2$ conformational transformations: (1) release of the fusion peptide from its pre-fusion pocket; (2) $HA_2$ extension; (3) insertion of the fusion peptide into the target membrane; (4) fold-back of the extended $HA_2$ intermediate (*Figure 1A*) (*Skehel and Wiley, 2000*). This last step brings together the fusion peptide

**eLife digest** Influenza is caused by viruses that infect birds and mammals. These viruses enter cells when two lipid bilayers—one surrounding the virus, the other enclosing the cellular compartment into which the virus has been engulfed—merge to form a single unified membrane. This process, known as membrane fusion, allows the RNA of the virus to gain access to the host cell's molecular machinery, which it commandeers to produce multiple copies of itself and to direct the assembly of new virus particles. The process of membrane fusion generally includes an intermediate hemifused state in which only the adjacent monolayers from each bilayer have merged. In addition to its role in virology, membrane fusion is critical for many other biological processes, including exocytosis, protein trafficking and the fertilization of eggs by sperm.

Efficient membrane fusion requires a catalyst, and a glycoprotein known as the influenza hemagglutinin performs this role for the influenza virus. The hemagglutinin is found on the surface of the virus, and a typical influenza virus particle can have a few hundred such molecules on its surface. When an influenza virus particle binds to the surface of a cell (as a result of these hemagglutinin molecules interacting with cellular receptor molecules), the cell engulfs the virus into an internal compartment called an endosome. Acidification of the endosome, part of the cell's normal activity, triggers a sequence of conformational changes in the hemagglutinin molecules on the surface of the virus. One part of the hemagglutinin inserts itself into the endosomal membrane, and further conformational changes draw the endosomal and viral membranes together into an intermediate, hemifused state; the process then continues until fusion of the two membranes is complete.

Previous work has suggested that an average of three hemagglutinin molecules are required to fuse the endosomal and viral membranes. Ivanovic et al. have now investigated the molecular details of this process and described the time course of conformational changes undergone by the hemagglutinin molecules from the moment the pH is lowered within the endosome until the time when hemifusion of the endosomal and viral membranes is complete. They find, among other things, that hemifusion proceeds rapidly only when three or four immediately adjacent hemagglutinin molecules have inserted into the endosomal membrane. Since membrane fusion is a very general cellular process, the findings of Ivanovic et al. are relevant to many areas of cell biology, in addition to having potential applications in virology.

and the C-terminal, transmembrane segment of each $HA_2$, anchored respectively in the viral and target membranes, which thus approach each other, either as apposed protrusions or as a single, target-membrane protrusion (*Kuzmin et al., 2001*; *Lee, 2010*). Fusion then ensues, initially as hemifusion (merger of only the proximal membrane leaflets) and then as formation of a continuous aqueous channel.

We report here a series of experiments that probe the relationship between HA structural properties and kinetic intermediates in the fusion pathway. *Floyd et al. (2008)* devised a method to monitor in real time the fusion of individual influenza virus particles with planar bilayers. Their results led to the conclusion that fusion requires on average three HA trimers, each of which independently undergoes the same, rate-limiting rearrangement, but they left undetermined the relationship between this conclusion and the inferred intermediates in *Figure 1A*. In the experiments reported here, we correlate HA structure with observed variations in fusion kinetics by comparing rates for appropriate HA mutants. We conclude that irreversible engagement of fusion peptides from 3–4 neighboring $[HA_1\text{-}HA_2]_3$ trimers, within a much larger virus-target-membrane interface, leads to subsequent rearrangements that rapidly and cooperatively induce membrane merger. Release of the fusion peptide from its pocket is rate-limiting for membrane engagement. A long-lived membrane-inserted extended intermediate is a fundamental aspect of the fusion mechanism.

## Results

### Hemifusion times and particle arrest

We recorded in real time a large number of individual influenza virions (approximately 1000 virions per field of view), labeled with a lipophilic fluorophore (R18), as they fused with a supported planar bilayer (see 'Materials and methods'). We followed hemifusion as a spike in individual virion fluorescence

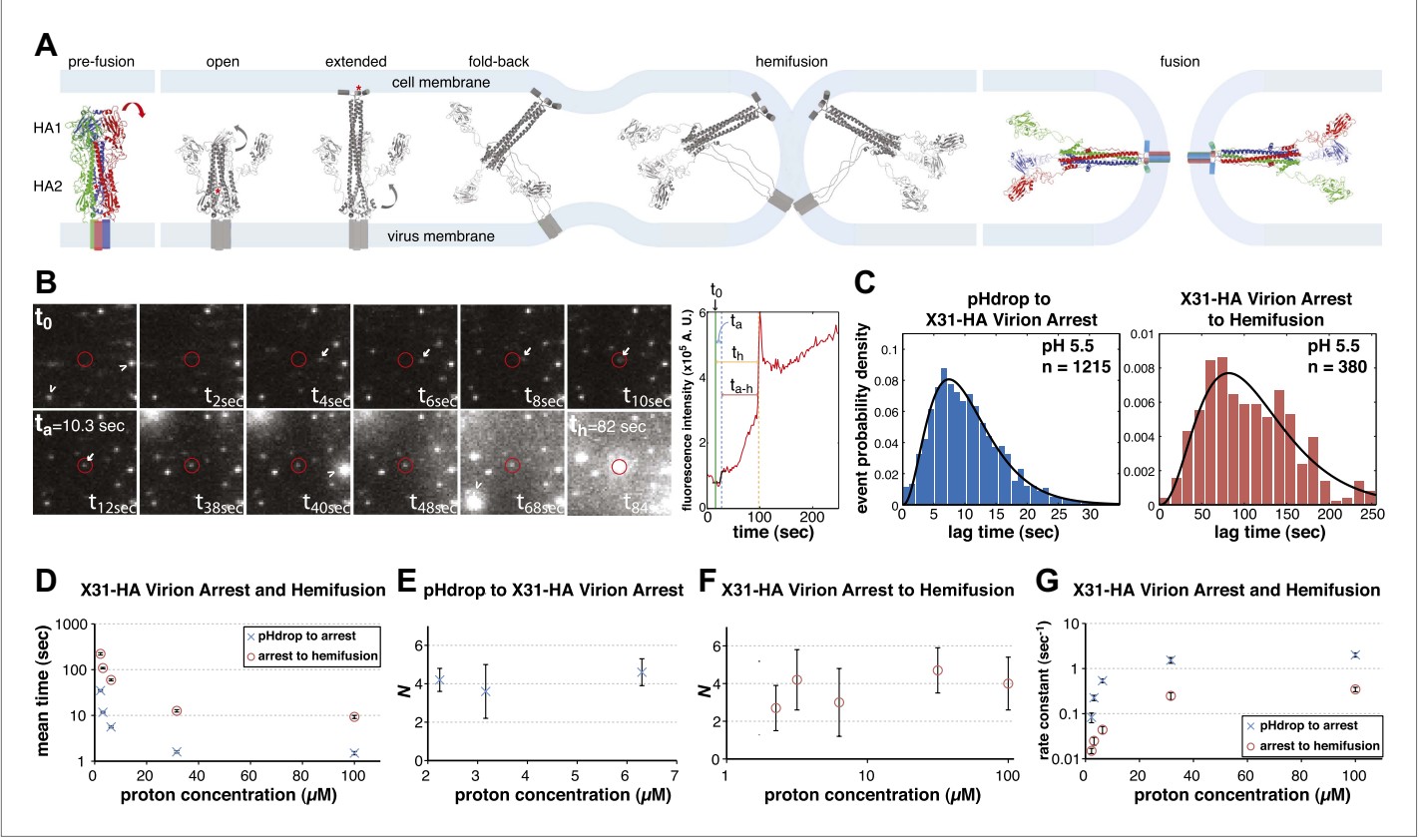

**Figure 1**. Single-virion analysis of fusion-promoting conformational change in influenza virus HA. (**A**) Hydrophobic fusion peptide (red asterisk) is initially inserted into a pocket near the trimer threefold axis. HA assumes an 'open' conformation upon proton binding allowing fusion-peptide release and membrane insertion. The fold-back of the extended intermediate causes hemifusion. The known pre-fusion and post-fusion HA structures are colored, and the inferred structural transitions are showed in gray. (**B**) *Left*: Tile view from pH drop ($t_0$) of a virion initially displaying directed motion (white arrow) followed by arrest ($t_a$) then hemifusion ($t_h$). Red circle marks the final virion location. Arrowheads mark two virions that were arrested at pH drop and hemifused at or just before $t_{40 s}$ and $t_{68 s}$ respectively. *Right*: Fluorescence trace of the virion circled in (**A**) (red line), line fitting the timing of virion arrest at its target location (black line) and parameters $t_0$ (green vertical line), $t_a$ (blue horizontal line) and $t_h$ (orange horizontal line) and arrest to hemifusion, $t_{a-h}$ (dark orange horizontal line). (**C**) $t_{lag}^{(pHdrop-arrest)}$ for all virions for which arrest values could be derived (*left*) and $t_{lag}^{(arrest-hemifusion)}$ for all virions for which both arrest and hemifusion values could be derived (*right*) with gamma-distribution fit (black line). Data are pooled from three independent experiments. (**D**) Mean $t_{lag}^{(pHdrop-arrest)}$ and $t_{lag}^{(arrest-hemifusion)}$. Error bars represent the standard error of the mean. (**E**) and (**F**) $N$ derived from fitting $t_{lag}^{(pHdrop-arrest)}$ (**E**) and $t_{lag}^{(arrest-hemifusion)}$ (**F**) with gamma probability density. (**G**) Rate constants derived from fitting $t_{lag}^{(arrest-hemifusion)}$ and $t_{lag}^{(pHdrop-arrest)}$ with gamma probability density and keeping $N$ fixed ($N = 3$) for both processes. (**B–G**) X31-HA virions have X31 HA in otherwise Udorn genetic background. Data shown are from representative experiments performed on the same day (n = 50 to 150) unless indicated that multiple experiments were pooled. Error bars represent 95% confidence interval for gamma fit-derived values unless otherwise indicated. Please refer to **Figure 1—figure supplement 1** for all histogram and gamma-distribution fit data plotted in (**D–G**).

The following figure supplements are available for figure 1:

**Figure supplement 1**. Histograms of $t_{lag}^{(pHdrop-arrest)}$ (*left*) and $t_{lag}^{(arrest-hemifusion)}$ (*right*) for X31-HA virions.

resulting from R18 fluorescence dequenching upon dilution into the target membrane (**Figure 1B**). We also observed a previously undetected fusion intermediate. Upon initiating the flow of buffer for pH exchange, most virions started to move under the hydrodynamic force while retaining contacts with the target bilayer. The particles arrested at various times after the pH drop, but invariably preceding hemifusion (**Figure 1B**; **Video 1**). The arrest was irreversible (**Videos 2 and 3**). A subset (approximately 20% at pH 5.5 and 5.65) of virions had already arrested at the onset of imaging or had done so before the pH drop. For lower final pH, a larger fraction had arrested by the time the pH drop was complete (**Table 1**). We determined lag times between pH drop and individual virion arrest ($t_{lag}^{(pHdrop-arrest)}$) and between arrest and hemifusion ($t_{lag}^{(arrest-hemifusion)}$) whenever both values could be extracted from the data (**Figure 1B,C**), for a range of proton concentrations between 2 and 100 μM (pH 5.65–4) (**Figure 1D**). Mean values

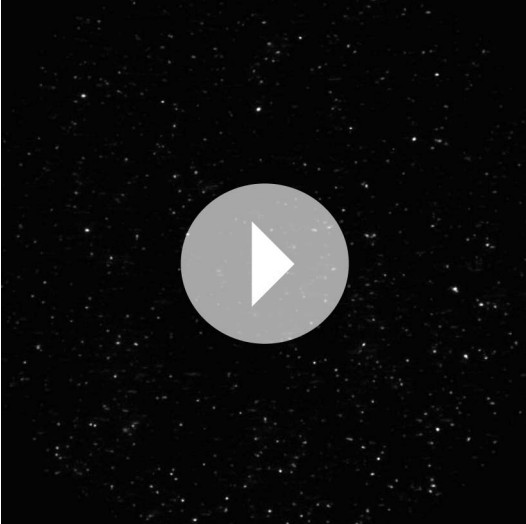

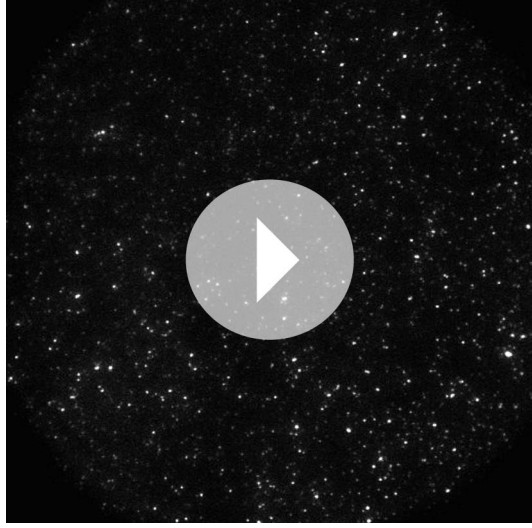

**Video 1**. X31-HA WT virion hemifusion at pH 5.5 from $t_0$ to $t_{230\,s}$ at 20× the actual rate.

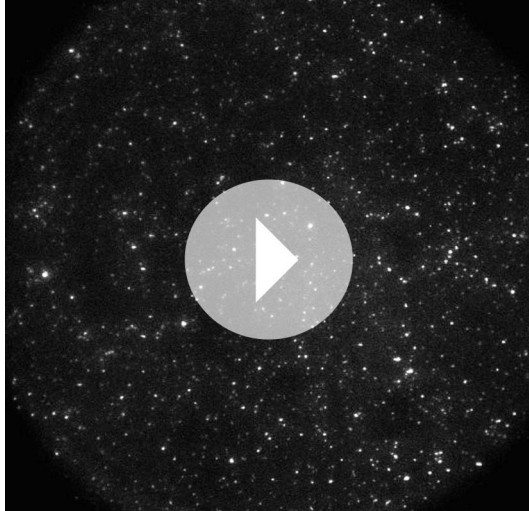

**Video 2**. Virion arrest is an irreversible intermediate of fusion. X31-HA WT virions were imaged. pH was dropped from 7.4 to 5.65 then brought back up to 7.4. Buffer flow was kept constant except when it was stopped to allow for the source buffer exchange back to neutral (between approximately $t_{45\,s}$ and $t_{95\,s}$).

**Video 3**. Virion arrest is an irreversible intermediate of fusion. A different field of view of the same experimental lane used in **Video 2** after the events imaged in **Video 2**. pH was dropped to pH 5.65. There is a marked reduction in the mobile fraction in **Video 3** relative to early times shown in **Video 2**. Furthermore, prearrested virions proceeded to hemifusion despite the intermediate reneutralization step. The videos are shown at 20× the actual rate.

for $t_{lag}^{(pHdrop–arrest)}$ and $t_{lag}^{(arrest–hemifusion)}$ show the same pH dependence, with mean $t_{lag}^{(pHdrop–arrest)}$ being about an order of magnitude shorter throughout the tested pH range.

We can fit the distributions for both $t_{lag}^{(pHdrop–arrest)}$ and $t_{lag}^{(arrest–hemifusion)}$ with one describing a requirement for $N$ independent events (either parallel or sequential; **Feller, 1968**), each with rate constant, $k$ (**Figure 1C** and **Table 1**). For any given particle, the two lag times are uncorrelated, ruling out any mutual dependence on overall particle properties such as length (**Figure 2**). We have further controlled for virion length by using only shorter-virion fractions in our experiments (see 'Materials and methods'). From the distributions, we derive $N = 3–4$ both for the number of events required to arrest virions following pH drop (**Figure 1E** and **Table 1**) and for the number of events required for hemifusion of the arrested particle (**Figure 1F** and **Table 1**) (see 'Materials and methods'); the latter is similar to the value obtained previously from $t_{lag}^{(pHdrop–hemifusion)}$ distributions (**Floyd et al., 2008**).

A simple interpretation of virion arrest is that insertion of fusion peptides from a number of independent HA trimers (3 or 4) into the target membrane anchors the particle. Bulk experiments have shown that short incubations of virions with target membranes at pH 5 and 0°C lead to insertion of a small subset of fusion peptides and stable virus anchoring not associated with membrane fusion (**Tsurudome et al., 1992**). We attribute the majority of pre-arrested events during high pH experiments (pH 5.5 and above) to imperfections in the bilayer or defective virions; the observed

**Table 1.** Arrest and hemifusion statistics for X31-HA WT and arrest statistics for S4G$_{HA2}^{Udorn}$ virions

| | Virion arrest to hemifusion* | pH drop to virion arrest† | |
|---|---|---|---|
| | **pH 5.5** | | |
| | X31-HA WT‡ | X31-HA WT‡ | S4G$_{HA2}^{Udorn}$§ |
| Number of virions | 380 | 1215 | 401 |
| Mean lag time (s)# | 105.1 ± 2.7 | 10.3 ± 0.2 | 9 ± 0.4 |
| N¶ | 3.7 ± 0.6 | 3.4 ± 0.2 | 3.4 ± 0.3 |
| Rate constant (s⁻¹) ¶ | 0.032 ± 0.006 | 0.33 ± 0.02 | 0.37 ± 0.03 |
| Mobile at pH drop** | | 81% | 76% |
| Mobile that hemifused** | | 75% | 65% |
| Static that hemifused** | | 82% | 81% |
| | **pH 4.5** | | |
| Number of virions | 155 | 392 | 385 |
| Mean lag time (s) # | 13.3 ± 0.6 | 1.4 ± 0.004 | 1.4 ± 0.004 |
| Mobile at pH drop** | | 46% | 38% |

*$t_{lag}^{(arrest–hemifusion)}$ for all virions for which both arrest and hemifusion values could be derived.

†$t_{lag}^{(pHdrop–arrest)}$ for all mobile virions for which arrest values could be derived.

‡X31-HA WT data are pooled from three independent experiments.

§S4G$_{HA2}^{Udorn}$ data are pooled from two independent experiments.

#Errors represent the standard error of the mean.

¶Errors represent the 95% confidence interval for the values derived from gamma-probability fits shown in **Figure 1C** (X31-HA virions) or **Figure 4C** (S4G$_{HA2}^{Udorn}$ virions).

**Percentages are derived from entire data sets.

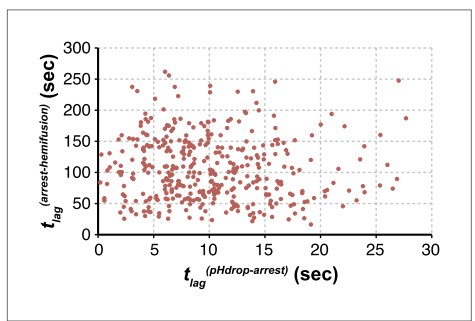

**Figure 2**. Virion arrest and hemifusion lag times are uncorrelated. $t_{lag}^{(arrest–hemifusion)}$ vs $t_{lag}^{(pHdrop–arrest)}$ for X31-HA virions at pH 5.5 (n = 380). Data are pooled from three independent experiments also shown in **Figure 1C**, **Table 1** and **Figure 4A,B** (X31–HA).

increase in the immobile fraction at lower pH might result from genuine triggering and membrane insertion of HA fusion peptides upon proton binding during the pH transition (see 'Materials and methods').

## Site-directed HA mutations and the rate-limiting step for HA rearrangement

To probe the molecular mechanism of arrest and its relationship to the mechanism of hemifusion (for which membrane insertion of the fusion peptide is clearly critical), we generated recombinant virus particles with site-directed mutations in HA. We used a set of plasmids derived from the A/Udorn/72 H3 influenza strain. The HA of Udorn is 97% identical in amino-acid sequence to that of X-31, the virus used in previous experiments (*Floyd et al., 2008*) and also the source of HA in otherwise Udorn genetic background in the experiments in **Figure 1**. Nonetheless, recombinant virions with Udorn HA had shorter hemifusion lag times in the physiologically relevant pH regime (pH > 4.5) than did those with X-31 HA (**Figure 3A**, **Videos 1 and 4**). Moreover, the Udorn-HA particles did not show directed motion under flow at any observable time point—that is, they had arrested by the time the low-pH transition was complete (see 'Materials and methods'). The Udorn hemifusion lag time ($t_{lag}^{(pHdrop–hemifusion)}$) distribution between pH 5.65 and 4 gave a pH-independent value of N close to 3 (**Figure 3B**), showing that differences between Udorn and X31 HAs do not affect the number of rate-limiting steps between virion arrest and hemifusion, but only their individual rate constants (**Figure 3C**).

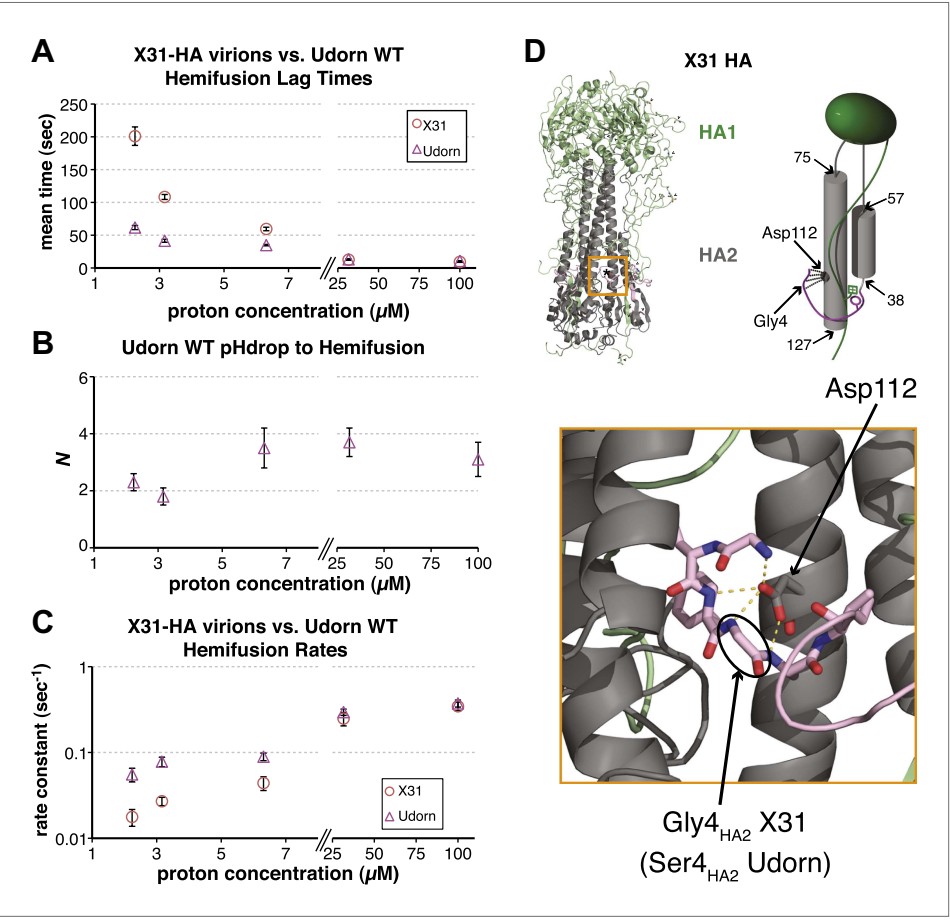

**Figure 3**. Udorn virions have accelerated hemifusion kinetics. (**A**) Mean $t_{lag}^{(pHdrop-hemifusion)}$ for Udorn and mean $t_{lag}^{(arrest-hemifusion)}$ for X31-HA virions. Error bars represent the standard error of the mean. (**B**) $N$ derived from fitting $t_{lag}^{(pHdrop-hemifusion)}$ with gamma probability density. (**C**) Rate constants derived from fitting $t_{lag}^{(pHdrop-arrest)}$ for Udorn and $t_{lag}^{(arrest-hemifusion)}$ for X31-HA virions with gamma probability density and keeping each $N$ fixed ($N = 3$). (**B–C**) Error bars represent 95% confidence interval for gamma fit-derived values. (**A–C**) Data shown are from representative experiments performed on the same day (n = 50 to 350). Please refer to **Figure 3—figure supplement 1** for all histogram and gamma-distribution fit data plotted in (**A–C**). (**D**) *Top left*: Ribbon representation of X31-HA trimer (**Weis et al., 1990**) showing positions of all residues that differ in Udorn-HA (arrowhead) including Gly4$_{HA2}$ (asterisk). *Top right*: Cartoon of an HA monomer emphasizing Asp112$_{HA2}$-fusion peptide hydrogen bond network and showing positions of residues along the HA$_2$ chain. *Bottom*: Close-up of the Asp112$_{HA2}$-fusion peptide network (region marked with an orange square on top left).

The following figure supplements are available for figure 3:

**Figure supplement 1**. Histograms of $t_{lag}^{(arrest-hemifusion)}$ for X31-HA virions (*left*) and of $t_{lag}^{(pHdrop-hemifusion)}$ for Udorn WT virions (*right*).

Comparison of the Udorn and X-31 HA amino-acid sequences suggests that one particular difference might account for the accelerated fusion kinetics of the former—a substitution of serine for glycine at position four in HA$_2$. Udorn is the only strain in the database with this substitution, glycine being otherwise universally conserved (**Nobusawa et al., 1991**; **Cross et al., 2009**). In the pre-fusion conformation of HA, Gly4$_{HA2}$ participates in a network of polar hydrogen-bond interactions with the carboxylate of Asp112$_{HA2}$, which is also strictly conserved (**Figure 3D**) (**Wilson et al., 1981**; **Weis et al., 1990**; **Russell et al., 2004**). A serine substitution would weaken or interrupt this interaction, because the glycine has a backbone conformation not allowed for other residues. Previous studies have reported an elevated pH threshold of fusion for cell-surface expressed HA harboring substitutions at Gly4 or Asp112 (**Gething et al., 1986**; **Steinhauer et al., 1995**). Gly4$_{HA2}$ might also favor a tight conformation

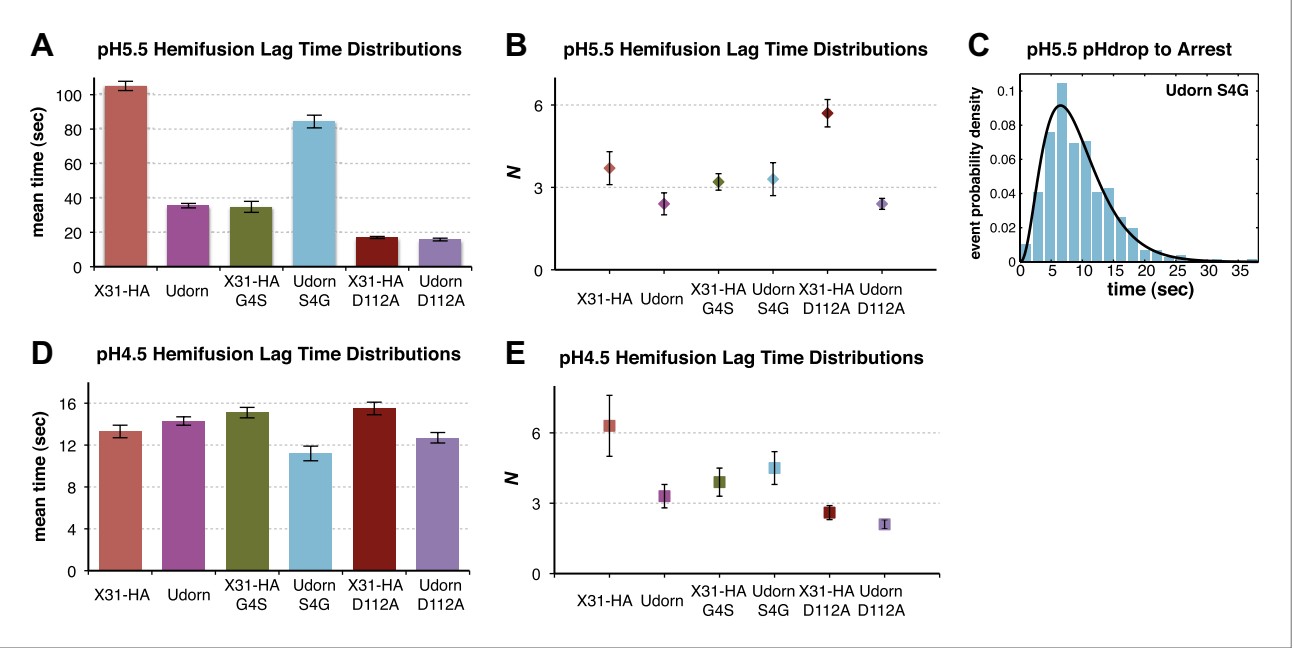

**Figure 4**. Fusion-peptide release from its pre-fusion pocket is a rate-limiting fusion-inducing molecular rearrangement in the physiologically relevant pH regime. (**A**) pH 5.5, mean $t_{lag}^{(arrest–hemifusion)}$ for X31-HA WT, S4G$_{HA2}^{Udorn}$ and mean $t_{lag}^{(pHdrop–hemifusion)}$ for Udorn WT, G4S$_{HA2}^{X31}$, D112A$_{HA2}^{Udorn}$ and D112A$_{HA2}^{X31}$. (**B**) $N$ derived from gamma-probability-density fits of the data analyzed also in (**A**). (**C**) Histogram of $t_{lag}^{(arrest–hemifusion)}$ distribution for S4G$_{HA2}^{Udorn}$ virions at pH 5.5 with the gamma-probability-density fit (black line). (**D**) pH 4.5, mean $t_{lag}^{(pHdrop–hemifusion)}$ for indicated virions. (**E**) $N$ derived from gamma-probability-density fits of the data analyzed also in (**D**). (**A–E**) Data shown are from pooled independent experiments (3, for Udorn and X31-HA WT virions, and 2, for Udorn and X31-HA mutant virions; n = 150 to 900). Error bars represent the standard error of the mean (**A** and **D**) or the 95% confidence interval for gamma fit-derived values (**B** and **E**). Please refer to **Figure 4—figure supplement 1** for all histogram and gamma-distribution fit data plotted in (**A**, **B**, **D** and **E**).

The following figure supplements are available for figure 4:

**Figure supplement 1**. Histograms of $t_{lag}^{(arrest–hemifusion)}$ distributions for X31-HA WT, S4G$_{HA2}^{Udorn}$ virions and of $t_{lag}^{(pHdrop–hemifusion)}$ for Udorn WT, G4S$_{HA2}^{X31}$, D112A$_{HA2}^{Udorn}$ and D112A$_{HA2}^{X31}$ at pH 5.5 (*top*).

of the membrane-inserted fusion-peptide helices (*Lorieau et al., 2010*), but its conservation does not seem critical for free fusion-peptide insertion into membranes (*Wharton et al., 1988*).

We generated a series of HA mutants in the context of intact, recombinant virions, to test the consequences for fusion kinetics of the Gly4$_{HA2}$–Asp112$_{HA2}$ interaction and of the fusion-peptide sequestration in activated HA. These included G4S$_{HA2}^{X31}$ and S4G$_{HA2}^{Udorn}$, as well as D112A$_{HA2}$ in both backgrounds. *Figure 4A,B* and *Videos 1 and 4–8* summarize our analysis of hemifusion rates for these variants at pH 5.5. The mutation G4S$_{HA2}^{X31}$ indeed accelerates hemifusion, and the reverse mutation S4G$_{HA2}^{Udorn}$ retards it (*Figure 4A*). Moreover, like WT Udorn virions, G4S$_{HA2}^{X31}$ virions do not show flow-induced motion at pH drop, while 76% S4G$_{HA2}^{Udorn}$ virions do (*Videos 5 and 6*, *Table 1*). Kinetic parameters for arrest, derived from the $t_{lag}^{(pHdrop–arrest)}$ distribution for the S4G$_{HA2}^{Udorn}$ particles are indistinguishable from those for X31-HA WT (*Figure 4C* and *Table 1*). These observations show that a single mutation, and thus probably the same molecular process, affects both the rate of arrest and the rate of hemifusion.

Data from D112A$_{HA2}$ mutants confirm that the Gly4$_{HA2}$–Asp112$_{HA2}$ interaction has an important influence on hemifusion rate at pH 5.5. D112A$_{HA2}$ mutations accelerate hemifusion of both Udorn and X31-HA virions, but the effect on the latter is more pronounced (*Figure 4A*). We infer from these observations that the G4S$_{HA2}$ mutation partly destabilizes contacts between the terminal fusion-peptide residues and Asp112$_{HA2}$ and that the D112A$_{HA2}$ change then eliminates this source of stability completely. None of the mutations change the values of $N$ derived from the pH 5.5 $t_{lag}^{(arrest–hemifusion)}$ distributions (or for static virions, $t_{lag}^{(pHdrop–hemifusion)}$ distributions) (*Figure 4B*), showing that these changes do not affect the number of steps. We conclude that for these viruses, release of the fusion peptide from its pre-fusion pocket is a rate-limiting step in the transition from arrest to hemifusion. The consistent changes in fusion

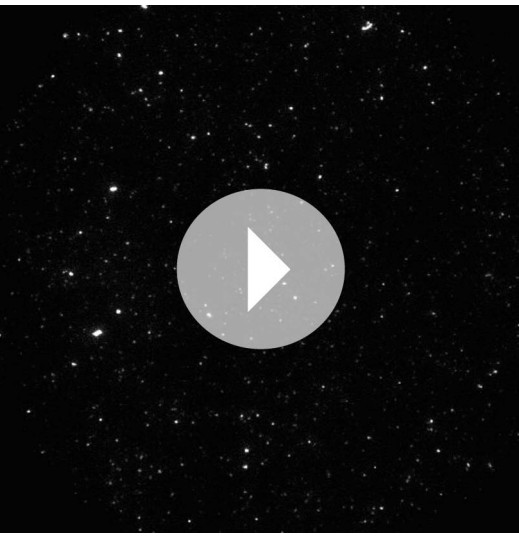

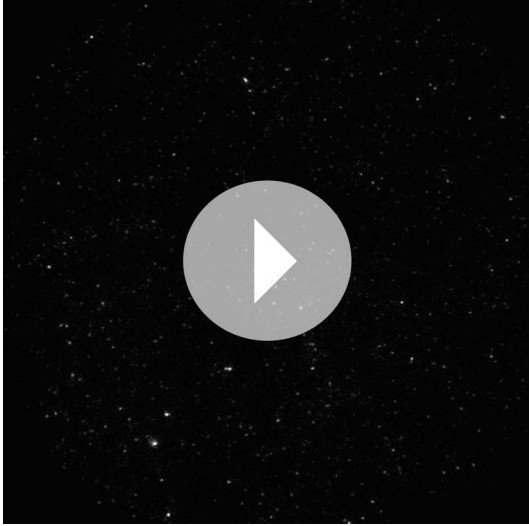

**Video 4**. Udorn WT virion hemifusion at pH 5.5 from $t_0$ to $t_{90\,s}$ at 20× the actual rate.

**Video 5**. G4S$_{HA2}^{X31}$ virion hemifusion at pH 5.5 from $t_0$ to $t_{90\,s}$ at 20× the actual rate.

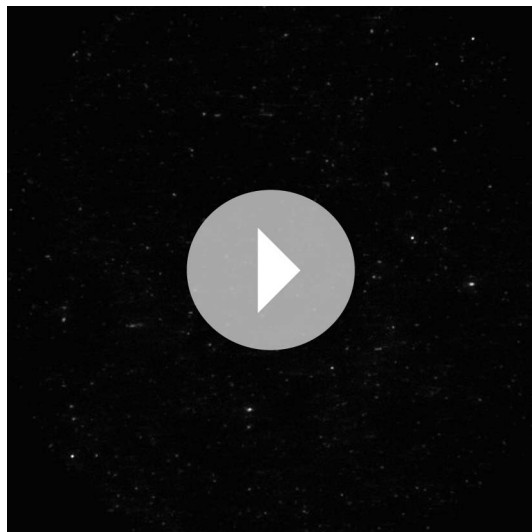

**Video 6**. S4G$_{HA2}^{Udorn}$ virion hemifusion at pH 5.5 from $t_0$ to $t_{180\,s}$ at 20× the actual rate.

kinetics produced by the mutations, including the good fit obtained from a probability distribution that assumes concomitant changes in the rates of all $N$ events (see 'Materials and methods'), further support our proposal that $N$ is related to the number of HA trimers that must independently expose their fusion peptides before a hemifusion-inducing transition can follow.

The rate of arrest for either of the destabilized D112A$_{HA2}$ variants was too rapid for us to detect flow-induced motion (***Videos 7 and 8***). We therefore could not determine $N$ from arrest distributions for those viruses, but we could nonetheless infer that fusion-peptide release from its pocket is rate-limiting for virion arrest as well as for hemifusion.

At pH 4.5 and below, the mutants described above have indistinguishable mean values for $t_{lag}^{(pHdrop-hemifusion)}$ (***Figure 4D***). Moreover, the lag times have become nearly pH independent, presumably because most of the relevant titrating groups (carboxylates) have become protonated. A subset (smaller than at pH 5.5) of X31-HA WT

and S4G$_{HA2}^{Udorn}$ virions exhibit directed motion under the force of buffer flow at the pH drop, both with mean times of just under 1.5 s (***Table 1***). Fitting the $t_{lag}^{(pHdrop-hemifusion)}$ distributions with a gamma probability density yields unchanged values of $N$ (between 3 and 6) for all the variants, confirming that what changes with pH is not the number of rate-limiting rearrangements leading to hemifusion but only their rate (***Figure 4E***). Protonation of Asp112 probably explains why mutations that affect stability of the fusion peptide in its pre-fusion pocket and influence the fusion rate at pH 5.5 have little or no effect on the fusion rate at pH 4.5 and below. The unchanged value of $N$ and the observation that a fraction of the X31-HA WT and S4G$_{HA2}^{Udorn}$ particles show directed motion after the pH drop suggest that even at pH 4.5, some rate-limiting rearrangement precedes membrane insertion of the fusion peptide.

The influence of mutations at positions 4 and 112 in HA$_2$ on hemifusion at the physiologically relevant pH of 5.5 demonstrates that the steps determining the rate of arrest (from the time of pH drop)

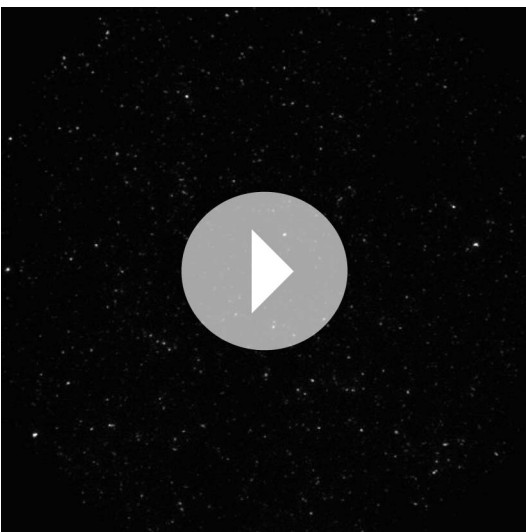

**Video 7**. D112A$_{HA2}^{X31}$ virion hemifusion at pH 5.5 from $t_0$ to $t_{40\,s}$ at 20× the actual rate.

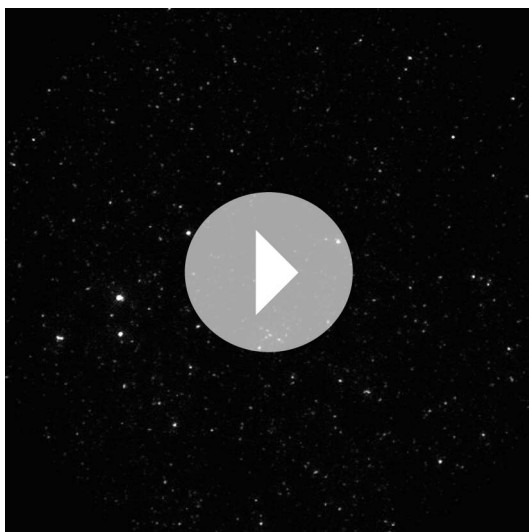

**Video 8**. D112A$_{HA2}^{Udorn}$ virion hemifusion at pH 5.5 from $t_0$ to $t_{35\,s}$ at 20× the actual rate.

and the rate of transition from arrest to hemifusion are the same, that is, both processes are limited by the rate of fusion-peptide exposure. Both also result from the action of several HA trimers each undergoing this rate-limiting rearrangement as indicated by the unchanged shape of their individual lag time distributions (unchanged value of $N$) under vastly different proton concentrations (**Figure 1**). The rate constants derived from gamma distribution fitting of the $t_{lag}^{(pHdrop-arrest)}$ data are, however, about five to tenfold greater than those for $t_{lag}^{(arrest-hemifusion)}$ (**Figure 1**), suggesting that the measured outcome in each case, arrest and hemifusion, is governed by different requirement(s). A simple interpretation of these observations is that <u>virion arrest</u> results from fusion-peptide release and irreversible target-membrane insertion from several HA trimers *anywhere* in the area of virus-target membrane contact and that <u>hemifusion</u> is limited by the same HA rearrangements, fusion-peptide release and membrane insertion, but requires a specific geometry for the participating HAs, that is, their proximity to each other, and is thus related to a different probability.

## Simulation of molecular events in the contact patch between virion and target membrane

To test the predictions of our model directly and to probe the relationship between the value of $N$ derived from gamma-distribution fitting of hemifusion lag times and the actual number of participating HA trimers in the hemifusion reaction, we developed the following computer simulation (**Figure 5**). We defined a contact patch of various sizes between 50 and 300 HAs as a circular area of hexagonally arranged trimers. Our lower-limit estimate of a contact area that can accommodate up to about 50 HA trimers, shown in **Figure 6**, is based on dimensions of a spherical virion with 55 nm in membrane-to-membrane diameter (**Calder et al., 2010**). Virions used in our current experiments are on average approximately 130 nm long and 55 nm wide (**Ivanovic et al., 2012**; see 'Materials and methods'), and thus the actual contact area can on average accommodate up to about 120 HA trimers. For each hypothetical virion we obtained lag times for individual HA triggering and membrane insertion events by random selection from an exponentially decaying function with a fixed rate, $k_{HApre-fusion-HAextended}$. We defined $t_{lag}^{(pHdrop-arrest)}$ as the time when the $N_a$th trimer within the contact area extended, and $t_{lag}^{(arrest-hemifusion)}$ as the time from virion arrest to the time when $N_h$th *neighbor* of the previously extended trimers extended. **Figure 5A** shows definitions of HA neighbors for $N_h$ values between 2 and 5 and accompanying simulation-derived arrest and hemifusion lag time distributions that assume $N_a = 3$ and a contact patch of 121 HA trimers.

Simulation-derived distributions agree remarkably well with our experimentally obtained data and predict $N_a = 3–4$, $N_h = 3–4$ and are most consistent with our independently predicted contact patch area of approximately 120 HAs (**Figure 5B**, *i* and *ii*). The gamma-distribution-derived value of $N$ for virion arrest closely approximates the number of trimers leading to arrest, $N_a$, but the apparent rate

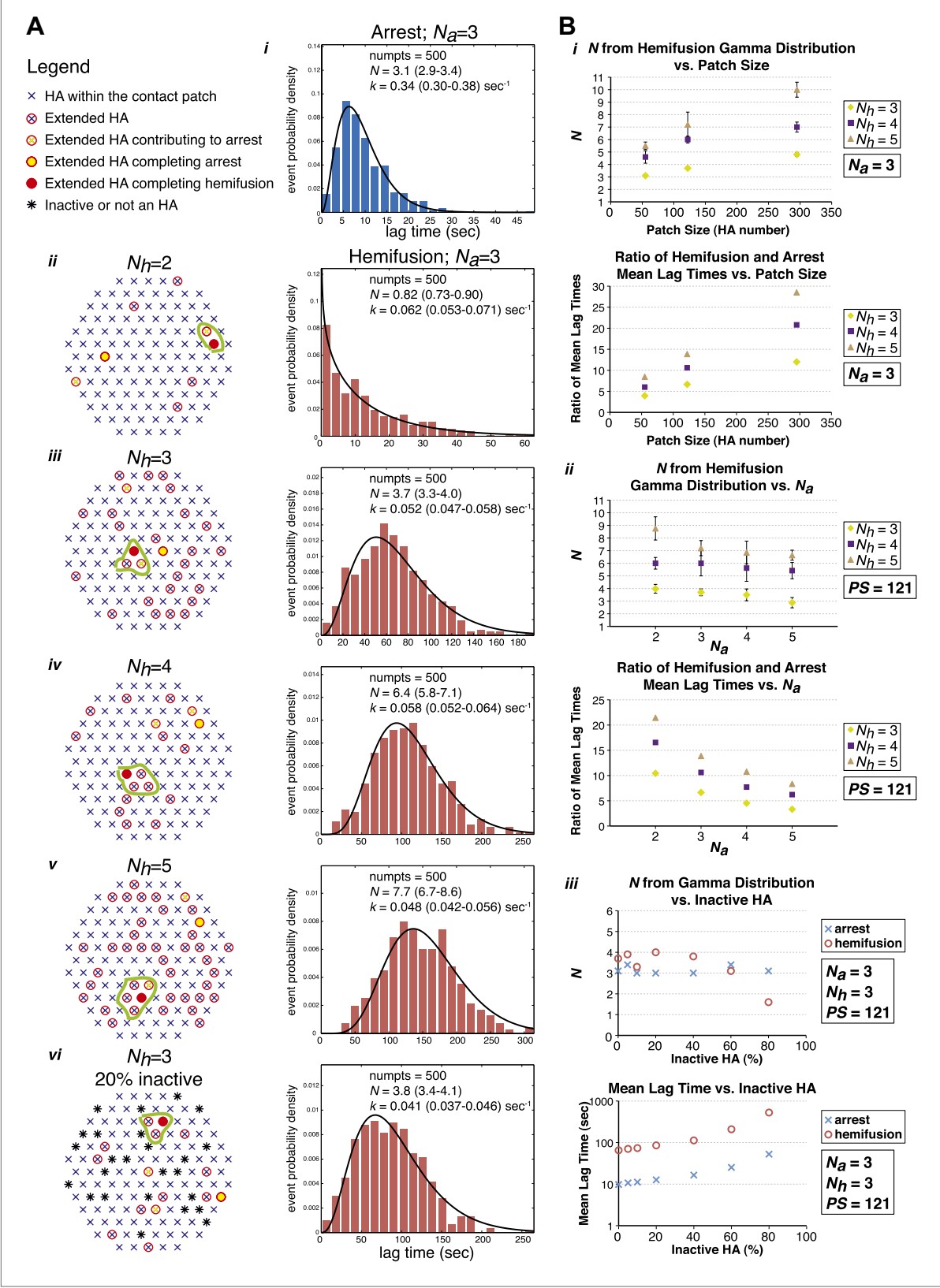

**Figure 5**. Computer simulation of the proposed model for HA-mediated membrane fusion. (**A**) *Left*: A contact patch containing 121 HA trimers and examples of individual virions and their contact-area HAs that have extended and inserted their fusion peptides into the target membrane by the time of *Figure 5. Continued on next page*

*Figure 5. Continued*

hemifusion, for simulated $N_h$ values of 2–5 and assuming either a contact patch consisting solely of active HAs (*ii–v*) or containing 20% inactive HAs randomly distributed throughout the contact area (*vi*). The neighboring HA clusters inducing hemifusion in each case are circled in green. *Right: $N_a$* = 3 and patch size = 121 containing either 0% or 20% inactive HAs as indicated in the corresponding diagrams; simulation-derived $t_{lag}^{(pHdrop–arrest)}$ distributions (*i*) and $t_{lag}^{(arrest–hemifusion)}$ distributions for $N_h$ = 2–5 (*ii–vi*) along with gamma-distribution fits (black curves) and fit-derived values for $N$ and $k$. Parentheses indicate 95% confidence interval for each fit-derived value. We collected data for 500 virions in each example. (**B**) (*i*) $N_a$ = 3; effect of patch size (PS) and $N_h$ either on gamma-fit-derived values for $N$ from hemifusion lag-time distributions (*top*) or on ratios of hemifusion and virion arrest mean lag times (*bottom*). (*ii*) Patch size = 121; effect of $N_a$ and $N_h$ either on gamma-fit-derived values for $N$ from hemifusion lag-time distributions (*top*) or on ratios of hemifusion and virion arrest mean lag times (*bottom*). (*iii*) $N_a$ = 3, $N_h$ = 3, patch size = 121; gamma-fit-derived $N$ values (*top*) and mean lag times (*bottom*) for arrest and hemifusion at a range of inactive HA concentrations. Data shown are averages, and error bars, when included, represent standard deviation for five independent simulations each performed for 500 virions. For MATLAB script and accompanying functions please refer to ***Source code 1***.

constant is substantially higher than the rate constant governing HA extension and membrane insertion (***Figure 5A***, *i*). The gamma-distribution-derived value of $N$ for hemifusion is related to the number of neighboring HA trimers participating in hemifusion, but it only approximates this value for smaller contact patch sizes (close to 50 HA trimers) and is greater than the true $N_h$ for larger patch sizes (***Figure 5B***, *i*). The value of $N_h$ = 2 yields a distribution that cannot be approximated by a gamma distribution for any analyzed patch size and value of $N_a$ (see ***Figure 5A***, *ii*, for one such example). Likewise, the value of $N_h$ = 5 yields a gamma distribution value of $N$ > 5 for all analyzed patch sizes and $N$ close to 7 for the predicted patch size of approximately 120 HA trimers (***Figure 5B***, *i*), inconsistent with our experimentally derived values. Our experimental data—the gamma-distribution-derived $N$ and the ratio of mean hemifusion and arrest times—are consistent with $N_h$ = 3–4 for patch sizes between 50 and 150 HA trimers. The apparent $N$-value-range between 3 and 6 (as observed for various mutants in ***Figure 4***) corresponds to the actual $N_h$ range between 3 and 4 and a patch size of about 120.

We have also addressed the dependence of the predicted data on any irregularities that might be present within the area of virus-target membrane contact, such as divergence from ideal hexagonal arrangement of HA trimers, presence of inactive HAs or intercalation of the viral neuraminidase (NA). We found that the key parameters—the gamma-distribution-derived $N$ value and the ratio of mean hemifusion and arrest lag times—are independent of the percentage of active HAs within the contact patch, down to a lattice-point occupancy of 40% (***Figure 5B***, *iii*). Further reduction in active HA density to 20% yielded a sudden drop in the apparent $N$ derived from gamma-distribution fitting of simulation-derived $t_{lag}^{(arrest–hemifusion)}$ data, consistent with previously published experimental evidence at pH 3, a proton concentration suggested to induce substantial HA denaturation (***Floyd et al., 2008***).

## Discussion

Many early studies of HA-mediated fusion kinetics relied on fusion of HA-expressing cells with red blood cells (***Spruce et al., 1989***; ***Zimmerberg et al., 1994***) or planar bilayers (***Melikyan et al., 1993a***, ***1993b***). In the latter case, measurements of conductance and capacitance showed reversible opening and closing of flickering pores, following initial acidification and commitment to secure pore opening only after a number of seconds. Good evidence that several HAs participate in pore formation came from red-blood cell fusion experiments in which the surface concentration of HA in the expressing cells varied over a roughly thirteenfold range (***Danieli et al., 1996***). Hill-plot treatment of the fusion times led to an estimate of 3–4 HAs per fusion event. A limitation of any cell–cell fusion experiment is the large contact zone and hence some ambiguity concerning the number of independent, local fusion events that occur in any one observation. Single-virion fusion experiments confirmed the earlier estimate of 3–4 HAs and showed that lipid-mixing precedes content mixing, as expected for an obligatory hemifusion intermediate (***Floyd et al., 2008***). In that work, the single, rate-limiting step for content mixing had a rate constant of about 0.05/s.

Calculations based on continuum models and molecular simulations have suggested various structures for hemifusion intermediates. The pathway with the lowest energy barrier appears to be through a so-called 'hemifusion stalk', derived from the apposed leaflets (***Kuzmin et al., 2001***). Some proposals for the subsequent transition to a fusion pore postulate expansion of the stalk into a 'hemifusion diaphragm', in which the distal leaflets form a continuous bilayer between the two membrane-bounded compartments; others suggest that a stalk can open directly into a pore. Hemifusion diaphragms can be seen by electron microscopy (e.g., ***Hernandez et al., 2012***); evidence for stalk-like structures

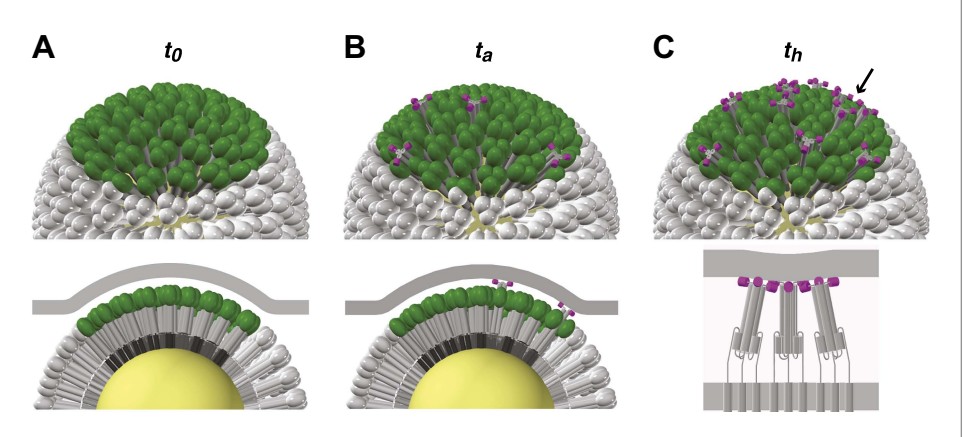

**Figure 6**. Stages of HA-mediated membrane fusion. Virion dimensions in the diagram are based on a spherical particle, 55 nm in membrane-to-membrane diameter (**Calder et al., 2010**). There are approximately 50 trimers in the contact area between the virus and the target cell, shown in green ($HA_1$) and gray ($HA_2$); the trimers outside of the contact area are shown in white. The contact area illustrated represents a lower limit, and the contact area for the somewhat elongated particles in our current experiments can likely accommodate up to approximately 120 HA trimers (see 'Results' and 'Materials and methods'). For clarity we present a virion surface containing only HA, but the conclusions we draw are independent of specific HA arrangement and potential irregularities in this region of contact (see **Figure 5**). We have also not included the target membrane in the perspective views (top), nor have we depicted the sialic-acid receptors in the cross-section views (bottom). The $t_0$, $t_a$ and $t_h$ labels are included to correlate the molecular transitions depicted in the model with the events defined in **Figure 1B**. (**A**) Virion engages a target membrane by HA-receptor interactions but can still move freely relative to the target-membrane area of contact. (**B**) Fusion peptides of several independent HA trimers have inserted into the target-membrane surface, immobilizing the virion on the target-cell surface. (**C**) Engagement of fusion peptides from three neighboring HA trimers allows subsequent rearrangements leading to hemifusion. $HA_1$s are left out for clarity from the bottom panel, which shows three neighboring trimers about to fold back.

comes from x-ray analysis of crystalline lipid phases (**Yang and Huang, 2002**). Both calculations and observations suggest that hemifusion diaphragms are stable, long-lived intermediates and that their formation might be a dead-end side reaction. Recent work on calcium-triggered SNARE/synaptotag-min/complexin-mediated fusion of lipid vesicles on a physiological time scale shows that the reaction proceeds too quickly (<100 ms) to detect even a hemifusion intermediate and that fusion from a hemi-fusion diaphragm is slow (tens of seconds) and inefficient (**Diao et al., 2012**). In influenza-virus fusion, tight packing of proteins on the viral surface probably makes a transition from stalk or other point-contact intermediate to diaphragm even less likely than in SNARE-mediated fusion, in which protein coverage of the fusing membranes is sparser.

The experimental results we describe above apply the previously developed single-virion experimental design (**Floyd et al., 2008**) to virus particles with site-directed mutations in the HA protein, enabling us to relate the kinetics of the lipid-mixing step to molecular events in the HA trimer inferred from the pre-fusion and post-fusion structures. The experiments report on conformational rearrangements in the catalyst (HA); their interpretation does not depend on the details of membrane organization in the substrates (the hemifusing membranes). They provide strong evidence for a long-lived, extended HA intermediate and for a rapid and cooperative transition to lipid mixing when a criti-cal number of these intermediates has accumulated within a contact neighborhood. Thus, they begin to elucidate features of the 'gray zone' in **Figure 1A**.

**Figure 6** illustrates the model derived from our current experiments (**Figures 1–4**), as sup-ported by our computer simulation (**Figure 5**). Binding to sialic-acid-containing glycans is weak, and local forces, such as flow, can displace the particle (either by 'rolling', with dissociation at the rear edge accompanied by new attachments at the front, or by translation across the membrane surface together with any associated lipids). The contact region between a virus particle and the target membrane is relatively large (probably 50–150 HA trimers, depending on the particle mor-phology: see comments above and **Figure 6** caption). Any fusion peptide released within this area

of contact and inserted into the target membrane will contribute to resisting buffer flow; the probability of arrest is then the probability for fusion-peptide exposure and insertion from 3–4 trimers within the area of membrane contact. Because hemifusion requires that fusion peptides of 3–4 adjacent trimers insert into the target membrane, the apparent rate constant for hemifusion relates to a different probability—that of inserting fusion peptides from several neighboring HAs. The data suggest that HA trimers independently undergo the first part of their low-pH-induced conformational change and pause as extended intermediates with their fusion peptides inserted into the target membrane. Capture by the target membrane restrains a trimer from collapsing to induce hemifusion until 2–3 neighbors help generate sufficient pull to deform the membrane. This interpretation also argues that when these conditions have been met, the final HA rearrangements are fast and cooperative.

Cooperativity of the final HA conformational rearrangements can derive either from contacts between neighboring HAs or simply from the linkage that is present among all participating HAs by virtue of their common insertion in the two hemifusing membranes. We favor the latter explanation. There is no evidence for defined, lateral contacts between HA trimers, and any such contacts, were they present, could not persist during the substantial changes in HA conformation and orientation that accompany the progression to hemifusion. The accumulation of $N$ extended $HA_2$ neighbors, each fluctuating toward a more stable, fully collapsed conformation while pulling against the elastic restoring force of the two membranes, will lead to an abrupt and cooperative event, when the critical $N$th molecule joins the cluster (rather like the abrupt consequence of adding a new member to the team on one side of a previously balanced tug-of-war). Evidence for the rapidity of the final HA rearrangements comes from the results of our simulation and gamma distribution analysis and the sensitivity of the derived rate constant to the mutations we have introduced. Any additional steps, not directly sensitive to those mutations, must be faster than any one of the steps of fusion-peptide engagement with the target membrane.

Engagement of the fusion peptide with a target membrane requires formation of an extended intermediate to project it beyond the outer margin of adjacent $HA_1$ heads (*Figure 6B*). For an intermediate to form that resembles the inner core of the ultimate, low-pH triggered structure (as drawn for the extended structure in *Figure 1A*), the $HA_1$ heads must dissociate from their contact with an $HA_2$ loop ($HA_2$ residues 58–74: see *Figure 3D*) between the short first α-helix ($HA_2$, residues 38–57) and the long central α-helix ($HA_2$, residues 75–127) (*Godley et al., 1992*), and that loop must in turn reverse direction and become itself a helix. Extension of the central trimeric coiled-coil probably drives the latter process, as the amino-acid sequence of the loop strongly favors such a conformation (*Carr and Kim, 1993*). Further necessary structural changes, in addition to release of the fusion peptide from its pocket, are dissociation of two strands ($HA_2$, residues 21–38) from the edge of the small β-sheet near the base of the HA trimer and displacement outward of this segment of $HA_2$ along with the short first helix, which could unfold and refold as it adds to the extending, central helical coiled-coil. The structure thus pictured would project the fusion peptide no more than about 25 Å beyond the palisade of un-triggered HAs (*Figures 1A and 6*), even if the $HA_2$ segment that will fold back against the central core remains roughly in its pre-fusion state (again, as drawn for the extended structure in *Figure 1A*). In short, the intermediate that mediates arrest and hemifusion must indeed be almost completely extended, at least for part of its lifetime.

The structures of the initial and final states do not dictate a unique ordering of the events just listed, but some must clearly precede others. Our results show that breaking the interactions that restrain the fusion peptide is rate-limiting both for virion arrest and for hemifusion at the pH of endosomes from which influenza virus penetrates (approximately 5 to 5.5); this interpretation is consistent with the dominance hierarchy of mutations that influence the threshold pH for fusion (*Steinhauer et al., 1996*). Mutations that affect the pH dependence of fusion map to a number of sites distributed across the $HA_2$ trimer interface (*Daniels et al., 1985*). Changes in the stability of this interface will influence fluctuations from the equilibrium structure and hence the 'window of opportunity' for the fusion peptide to withdraw productively from its pocket. Emergence of the fusion peptide can in principle precede the other events, which then must follow rapidly (or the peptide will reinsert); perhaps more frequently, extension of the central coiled-coil by residues 20–74 may exert a 'tug' to extract the fusion peptide. The fusion-peptide contacts, conserved in all field-isolated influenza HA virions, form upon cleavage of $HA_0$ (*Chen et al., 1998*; *Russell et al., 2004*). Their stability is critical for viral transmission, not only in the passage from one host or cell to another, but also during entry, as fusion will be relatively unproductive if it occurs before the pH of the surrounding compartment has dropped enough to allow M2-mediated acidification of the particle interior (*Ivanovic et al., 2012*), which induces dissociation of the eight

ribonucleoproteins (RNPs) of the viral genome from the matrix protein, M1 (*Martin and Helenius, 1991*). Fusion too rapid for adequate acidification would release an inactive M1-RNP complex.

Evidence for extended intermediates for other viral fusion proteins is largely indirect. The most extensive data are for HIV gp41, in which binding of peptides such as C34 (*Chan et al., 1998*) and T-20 (*Wild et al., 1994*), mimics for the outer layer of the final, hairpin structure, indicates that the inner-core coiled-coil must be present before the outer layer folds back against it. Outer-layer peptides from paramyxovirus F proteins similarly inhibit fusion (*Lambert et al., 1996*). Electron microscopy of human parainfluenza virus 5 (PIV5) fusing with target vesicles shows a distance between viral and target membranes that corresponds to the gap expected for the putative extended intermediate of F, which (unlike the HA intermediate) should project much farther from the viral membrane than does the pre-fusion form (*Kim et al., 2011*).

The relative kinetics of HA-mediated hemifusion and arrest show that because fusion requires several neighboring HA trimers, the lifetime of an isolated extended intermediate is substantial. Release of the fusion peptide is rate limiting for generating an extended HA intermediate, but subsequent collapse is rapid only when three or four neighboring extended intermediates have appeared, and the mean lifetime of the 'pioneer' extended HA in any patch can be as long as a minute or more, depending on pH. Thus, a relatively long-lived extended intermediate could prove to be a clinically useful target for an influenza-virus entry inhibitor: the inherent lower limit to the stability of the pre-fusion HA combined with the requirement for the coordinated action of multiple, independently-triggered HAs to induce fusion, curbs the ability of a virus to escape such treatment by mutations that accelerate fusion-inducing rearrangements.

The only clinically approved inhibitor of virus fusion targets the extended intermediate of the HIV fusion protein, gp41 (*Wild et al., 1994*; *Kilby and Enron, 2003*); analogous inhibitors have been designed for paramyxoviruses (*Lambert et al., 1996*), influenza virus (*Lee et al., 2011*) and flaviviruses (*Schmidt et al., 2010a*, *2010b*). For HIV fusion, the results of timing-of-inhibitor-addition experiments (*Gallo et al., 2001*; *Miyauchi et al., 2009*), the low surface density of the HIV-envelope protein (*Chertova et al., 2002*; *Zhu et al., 2003*), and the indication, from experiments with virions bearing mixtures of active and inactive spikes, that one HIV-envelope trimer may be sufficient to mediate fusion (*Yang et al., 2005*), have led to the inference that a single, trimeric gp41 mediates fusion through a very slow fold-back transition. The data presented here for influenza HA show that there can be a long-lived species, analogous to the extended form of gp41, even when the fold-back transition itself is fast. We suggest that the long delay between HIV binding and membrane fusion may reflect a sequence of events similar to those in our model for influenza virus fusion. In other words, we propose that HIV fusion may require recruitment of (at least) a second envelope trimer, following attachment of the first. The delay time would then be a combination of the time needed to recruit one or more additional envelope proteins and the time needed to activate them with CD4 and a coreceptor. Extensions of the approaches taken here to HIV and other viruses will be necessary to probe the true similarities and differences among them.

## Materials and methods

### Virions

We based the protocol for recombinant influenza rescue on that developed by *Neumann et al. (1999)* and later adapted for A/Udorn/72 by *Takeda et al. (2003)* and which we described previously (*Ivanovic et al., 2012*). All virions in the current study contained either Udorn or X31 HA, and the remaining genome segments were derived from A/Udorn/72. Virus plaque purification, passaging and purification of the third passage virus were performed as previously described (*Ivanovic et al., 2012*) with the following modifications. We passaged both $D112A_{HA2}$ mutants at reduced temperature (34°C instead of 37°C) and increased multiplicity of infection (0.05 PFU/ml instead of 0.001 PFU/ml). All final (third passage) infections were performed at 34°C. For the current experiments we used only the fractions enriched in 'short' virions (average membrane-to-membrane length approximately 130 nm) (we previously published details about fractionating virions by size on sucrose gradients; *Ivanovic et al., 2012*).

We introduced specific mutations in Udorn or X31 HAs by site-directed mutagenesis with QuickChange (Stratagene, La Jolla, CA) based on manufacturer's recommendations. We prepared two independent clones (derived from different plaques after initial virion rescue) for each WT and mutant viruses. We show data for only a single set of clones, but we reached the same basic conclusions with the second set: that $G4S_{HA2}^{X31}$ and $S4G_{HA2}^{Udorn}$ mutations reverse the arrest and hemifusion rate properties

and that D112A$_{HA2}$ mutations accelerate hemifusion in both Udorn and X31-HA backgrounds, but the effect on X31-HA virions is more pronounced.

We verified the presence of engineered mutations and the absence of spurious mutations in the open reading frame of the HA segment RNA of the purified viral products as follows. RNA was extracted using RNeasy mini kit (Qiagen, Valencia, CA), full-length HA segment RNA was reverse transcribed and amplified with universal primers specific for HA genome segment ends (*Hoffmann et al., 2001*) using One-step RT-PCR kit (Qiagen) and sequenced using the same set of primers. The G4S$_{HA2}$ mutation in Udorn HA$_2$ (GenBank: ABD79032.1), present also in the DNA used to derive Udorn HA by reverse genetics, is probably an anomaly of laboratory propagation at some point in the history of deriving the strain. Indeed, G4S$_{HA2}$ was one of two mutations that arose in one isolate after several passages of a recombinant virus in mammalian cells (*Lin et al., 1997*).

Completeness of HA cleavage/activation in the purified virus preps was verified by Western Blotting using anti-HA-tag mouse monoclonal antibody (Cell Signaling, Danvers, MA), anti-mouse HRP secondary antibody and ECL Plus Western Blotting Detection System (GE Healthcare, Little Chalfont, UK).

## Single-virion fusion

### Preparation of coverslips, flow cells and fluorescein-labeled planar bilayer

Glass coverslips were cleaned extensively by two rounds of sequential 20-min sonication in detergent, 1 M potassium hydroxide, acetone then ethanol. They were dried at 100°C for 30 min to 1 hr then plasma cleaned for 3 min at 500 mTorr O$_2$ and 75 W (March Plasmod Plasma Etcher; March Instruments, Inc., Concord, CA). The final sonication in ethanol and plasma cleaning steps were done immediately before each experiment.

PDMS flow-cell fabrication was described previously (*Ivanovic et al., 2012*). Flow cells were affixed to clean-glass coverslips prior to planar membrane bilayer preparation as described (*Floyd et al., 2008*). Liposomes used to generate planar bilayers consisted of 4:4:2:0.1:2 × 10$^{-4}$ ratio of 1, 2, dioleoyl-sn-glycero-3-phosphocholine (DOPC) (Avanti Polar Lipids, Alabaster, AL), 1-oleoyl-2-palmitoyl-sn-glycero-3-phosphocholine (POPC; Avanti Polar Lipids), cholesterol (Avanti Polar Lipids), bovine brain disialoganglioside GD1a (Sigma, St. Louis, MO), and N-((6-(biotinoyl)amino) hexanoyl)-1,2-dihexadecanoyl-sn-glycero-3-phosphoethanolamine(biotin-X DHPE) (Molecular Probes, Life Technologies, Grand Island, NY), where sialic acid residues on GD1a served as receptors for influenza virions. We extruded liposomes through a polycarbonate membrane filter with a 200-nm pore size, loaded liposomes into flow cells at 0.03 ml/min for 1 min, then stopped the flow and allowed 15 min for bilayers to form by vesicle-spreading method (*Nollert et al., 1995*). The bilayers were washed with neutral pH buffer (50 mM HepesNaOH pH 7.4, 137 mM NaCl and 0.2 mM EDTA) flowing at 0.05 ml/min for 2 min. They were labeled with fluorescent-labeled streptavidin (30 µg/ml; Invitrogen) for 5 min then washed again with neutral pH buffer.

### Single virion fusion

We used the protocol for influenza membrane fusion described previously (*Floyd et al., 2008*) with modifications. We labeled virions in 5- to 20-µl reactions with a lipophilic dye Octadecyl Rhodamine B Chloride (R18; Invitrogen, Life Technologies, Grand Island, NY) at a quenching concentration, 40 µM, for 2 hr at room temperature. Concentration of total viral protein in virion labeling reactions was between 0.25 and 1 mg/ml. Unincorporated dye was removed by gel filtration using PD-10 desalting column (GE Healthcare) according to manufacturer's recommendations using the neutral pH buffer, and three most concentrated, 250-µl fractions were pooled. Labeled virions were used within 48 hr of labeling. Virions were loaded into the flow cell at 0.05 ml/min for 30 s, and unattached virions were washed with neutral pH buffer within 2 min of introducing the virus into the flow cell. The pH in the flow cell was dropped at 7 min post virus introduction using citrate buffer (10 mM citric acid, 140 mM NaCl and 0.1 mM EDTA) having indicated pH. The flow rate of the low-pH buffer was kept constant at 0.075 ml/min for the duration of imaging, except in a single experiment (*Video 2*), where the flow was stopped to allow switching from low-pH back to neutral-pH buffer flow. All experiments were performed at 23°C.

### Microscope configuration

We used the microscope configuration described previously (*Floyd et al., 2008*) except that 488-nm laser (Obis, Coherent, Santa Clara, CA) used to excite fluorescein and 552-nm laser (Obis, Coherent)

used to excite R18 were used simultaneously and each was projected onto an entire 512 × 512 pixel EM-CCD sensor (Model C9100-13; Hamamatsu, Bridgewater, NJ). The power of the 488-nm laser was 1–2 µW and that of the 552-nm laser was 0.5–1.5 µW over the approximately 160 µm in diameter area. Acquisition times were between 0.5 and 2 s, depending on pH.

## Data analysis

We used MATLAB (MathWorks, Natick, MA) for all data analysis.

### Derivation of pH-drop time ($t_0$)

The reference time point ($t = 0$) for our kinetic measurements was the time at which most of the fluorescein signal had dissipated, determined as follows. We integrated the total optical fluorescence power in each imaged field from the onset of imaging to a time when at most a small subset of virions had hemifused. We generated fluorescence-vs-time plots of these data. We fitted the transition step resulting from fluorescein dissipation with the integral of a Gaussian function with a step-down shape, as previously described for dissipation of internal virion fluorescein (*Ivanovic et al., 2012*), using the following model:

$$f(t) = y_{ofs} + \frac{h}{2} erfc\left(\frac{t - t_c}{w}\right) e^{-m(t-t_c)},$$

where $h$ is the height of the intensity drop, $w$ is the half-width of the transition (in s), $t_c$ is the time at which the transition is half complete, $y_{ofs}$ is the residual intensity after dissipation, and $m$ is a decay rate (slower than $1/w$); $erfc(\cdot)$, the complementary error function, is the integral of a Gaussian, defined as

$$erfc(t) \equiv \frac{2}{\sqrt{\pi}} \int_t^\infty e^{-x^2} dx \text{ or } erfc(t) = 1 - erf(t),$$

(see below for the definition of $erf(t)$). We define $t_0$ as ($t_c + w$).

### Derivation of virion-arrest time ($t_a$)

Individual virion positions after their arrest or after the pH drop (for static virions) were detected automatically and their fluorescence trajectories extracted as previously described (*Floyd et al., 2008*). We fitted the local intensity jump due to virion arrival and arrest at this location (see *Figure 1B*) with the integral of a Gaussian with a step-up shape:

$$f(t) = y_{ofs} + \frac{h}{2}\left[1 + erf\left(\frac{t - t_c}{w}\right)\right] e^{-m(t-t_c)},$$

where $erf(\cdot)$, the error function, is defined as

$$erf(t) \equiv \frac{2}{\sqrt{\pi}} \int_0^t e^{-x^2} dx,$$

and the remaining parameters are as described above for the model used to fit fluorescein dissipation. We define $t_a$ as ($t_c + w$) − $t_0$.

### Derivation of hemifusion time ($t_h$)

Hemifusion time for individual virions was determined as previously described (*Floyd et al., 2008*) using $t_0$ described above as the reference point.

### Fitting of lag time data

We fitted $t_{lag}^{(pHdrop-arrest)}$, $t_{lag}^{(arrest-hemifusion)}$, $t_{lag}^{(pHdrop-hemifusion)}$ with a gamma distribution, $P_O(t) = \frac{k^N t^{N-1}}{\Gamma(N)} e^{-kt}$, to derive $k$ and/or $N$ (*Figures 1E,F, 2B, 3B,E* and *Table 1*), or to derive $k$ while keeping $N$ fixed (see below) (*Figures 1G, 2C*). The 95% confidence intervals for the fits were automatically generated.

Our current estimates for $N$ describing either arrest to hemifusion for X31-HA virions or pH drop to hemifusion for Udorn virions was between 3 and 4 (*Figures 1F, 2B* and *Table 1*), similar to previously published value of 3 for X31 virions (*Floyd et al., 2008*). All hemifusion $k$ values in *Figures 1G and 2C*

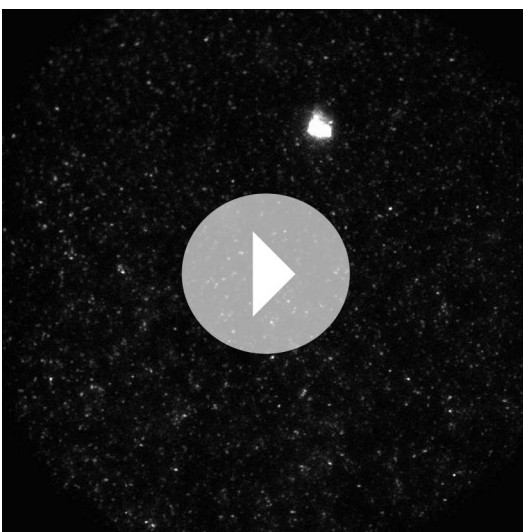

**Video 9**. X31-HA WT virions were imaged under constant, pH-7.4 buffer flow, starting at ~3 min post virus attachment. The video is shown at 10× the actual rate.

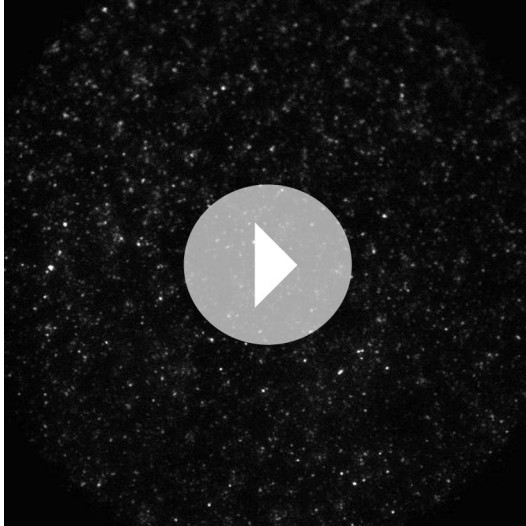

**Video 10**. A different field of view of the same experimental lane used in **Video 9**. X31-HA WT virions were imaged starting at ~7 min post virus attachment under constant buffer flow and pH was dropped to 5.5. pH drop can be observed at just under 2 s into the video as a drop in background fluorescence, concomitant with which virions accelerate. The video is shown at 10× the actual rate.

were derived from gamma-distribution fitting of $t_{lag}^{(arrest–hemifusion)}$ (X31-HA virions) or $t_{lag}^{(pHdrop–hemifusion)}$ (Udorn virions) distributions with fixed $N = 3$ and free-fitted $k$.

Because the time for the external pH to drop from neutral to a target value in our flow cell was significant relative to the average $t_{lag}^{(pHdrop–arrest)}$, especially at high proton concentrations (the duration of the pH transition was equal to the mean $t_{lag}^{(pHdrop–arrest)}$ at pH 4 and 4.5 and about one-tenth of that interval at pH 5.65), the true value of $N$ for pH drop to arrest is approached by experiments at low proton concentrations. We derive a value for $N$ of about four from the $t_{lag}^{(pHdrop–arrest)}$ distributions at low proton concentrations (2 to 6 µM or pH 5.65–5.2) (**Figure 1F**). We get a more accurate estimate for $N$ after pooling $t_{lag}^{(pHdrop–arrest)}$ data from three independent experiments at pH 5.5 (n = 1215) to obtain a value of 3.4 ± 0.2 (**Figure 1C**, *left*, and **Table 1**). All arrest $k$ values in **Figure 1G** were derived from gamma-distribution fitting of $t_{lag}^{(pHdrop–arrest)}$ distributions with fixed $N = 3$ and free-fitted $k$, and thus represent a lower limit estimate.

The efficiency of arrest for all virions at all analyzed pHs is close to 100%. For approximately 20% of X31-HA WT and $S4G_{HA2}^{Udorn}$ virions and nearly 100% of Udorn WT and all destabilized mutant virions ($G4S_{HA2}^{X31}$, $D112A_{HA2}^{Udorn}$ and $D112A_{HA2}^{X31}$), we failed to detect significant movement following pH drop, even in the higher pH range. In some cases, we cannot unambiguously distinguish between (1) virion arrest due to virion or bilayer defects, such as presence of virion-associated denatured HA or parts of the target bilayer that are not continuous with the rest of the membrane surface (e.g., remaining unfused liposomes), and (2) genuine HA triggering and membrane insertion that precede completion of the pH drop and might even occur with low frequency at neutral pH. One reason for this limitation is that virion movement under hydrodynamic force is somewhat retarded at neutral pH even in the context of X31-HA WT virions (**Video 9**); upon pH drop, X31-HA WT virions initially accelerate and then arrest (**Video 10**). HA triggering and membrane insertion at neutral pH is probably a rare event for all virions analyzed in this study but one that might occur more frequently for Udorn WT and other virions with a destabilized HA. Indeed, we occasionally observe hemifusion at neutral pH even with X31-HA WT virions (see **Video 9**).

We believe that the fraction (approximately 20%) of X31-HA WT and $S4G_{HA2}^{Udorn}$ virions that never move under hydrodynamic force at high pH mainly results from virion or bilayer defects and not from HA triggering at neutral pH for three reasons: (1) the observed fraction of immobile virions saturates above pH 5.5, while a modest increase in pH to 5.65 has a twofold to threefold effect on mean lag times for both arrest and hemifusion; (2) hemifusion kinetics are not significantly different for

pre-arrested and mobile populations, and a majority of virions (65% to 80%) from both populations undergo hemifusion (see *Table 1*); (3) the observed fraction of immobile virions varies to some extent (15–30%) even when mean arrest and hemifusion times are unchanged from experiment to experiment.

For Udorn WT and destabilized mutants, a combination of factors might explain the nearly complete absence of motion at pH drop: some fraction of HAs might release fusion peptides even at neutral pH (although probably not very frequently, as neutral pH fusion is still a rare event), and they might arrest before low pH-induced motion takes effect (see *Video 10* for low pH-induced motion of X31-HA WT virions).

## Computer simulation

We developed a computer simulation of our proposed model using MATLAB (MathWorks).

### Generate contact patch

We defined a hexagonal lattice of HA trimers, in which each HA has exactly six neighbors each at a distance $a$. We then defined a circle of diameter, $D = 2 \times \sqrt{\dfrac{n}{\pi}} \times a$, where $n$ is an approximate number of HA trimers in the contact patch, and then identified the positions and the total number of HAs within the circle, $n_{actual}$. For simulations involving a reduced fraction of active HAs, a given fraction of $n_{actual}$ HAs at random positions within a contact patch were flagged as inactive.

### HA-extension lag times, arrest and hemifusion

We assumed that for each HA trimer, its conversion to an extended, membrane-inserted form is a single-step process, governed by a rate constant, $k_{HApre-fusion-HAextended}$. We determined an arbitrary value for the extension rate constant of 0.0025/s and kept this value unchanged in all our simulations.

We first obtained lag times for extension of each HA trimer in a given patch by random selection from an exponentially decaying function with rate $k_{HApre-fusion–HAextended}$. We sorted lag times for all HAs in ascending order. $t_{lag}^{(pHdrop–arrest)}$ was defined as the lag time of $N_a$th *active* HA trimer in a contact patch (where $N_a$ is a predetermined number of HAs that results in virion arrest). $t_{lag}^{(arrest–hemifusion)}$ was defined as the lag time of that HA trimer that extended next to a defined number ($N_h − 1$) of *active neighbors* with shorter lag times minus the $t_{lag}^{(pHdrop–arrest)}$. $N_h$ is a value between 2 and 5 and is a predetermined number of extended *active neighbors* that results in hemifusion. Definitions of *neighbors* for $N_h$ values between 2 and 5 that were used in our simulations are outlined in *Figure 5A*. The process was repeated for 500 virions in a typical simulation experiment.

Simulation data for $t_{lag}^{(pHdrop–arrest)}$ and $t_{lag}^{(arrest–hemifusion)}$ were fitted as described under 'Data analysis' for experiment-derived data.

## Acknowledgements

We thank L'Oreal and AAAS for the 2011 Women in Science Fellowship to TI that enabled construction of a custom TIRF microscope used in the current experiments, Branislav Ivanovic for 3D drawings of HA monomer (*Figure 3*) and the model of influenza membrane fusion (*Figure 6*), and Milos Popovic for valuable input and discussion. Robert Garcea generously made his laboratory and equipment available to TI for this work.

## Additional information

### Funding

| Funder | Grant reference number | Author |
| --- | --- | --- |
| Howard Hughes Medical Institute | | Stephen C Harrison |
| National Institutes of Health | U54AI057159 | Stephen C Harrison |
| AAAS/L'Oreal | Women in Science 2011 | Tijana Ivanovic |
| National Institutes of Health | R21-AI072346 | Antoine M van Oijen |
| National Institutes of Health | AI081842 | Sean P Whelan |

The funders had no role in study design, data collection and interpretation, or the decision to submit the work for publication.

**Author contributions**

TI, Conception and design, Acquisition of data, Analysis and interpretation of data, Drafting or revising the article; JLC, Acquisition of data; SPW, Conception and design, Analysis and interpretation of data; AMvO, Conception and design, Analysis and interpretation of data, Drafting or revising the article; SCH, Conception and design, Analysis and interpretation of data, Drafting or revising the article

## Additional files

### Source code 1.

• Source code 1. Simulation script, s_arrest_hemifusion_simulation.m, and accompanying functions, generate_patch.m, s_randomdist.m, isaN2tuplet.m, findFlippedNeighbors.m, for MATLAB version R2012a.

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
