## [Decision Letter]

Thank you for choosing to send your work entitled “Influenza-virus membrane fusion: cooperative fold-back of stochastically induced hemagglutinin intermediates” for consideration at *eLife*. Your article has been evaluated by a Senior editor and 3 reviewers, one of whom is a member of our Board of Reviewing Editors.

The Reviewing editor and the other reviewers discussed their comments before we reached this decision, and the Reviewing editor has assembled the following comments based on the reviewers' reports.

## General assessment

This paper is an important contribution towards understanding the molecular mechanism of HA fusion. Harrison, van Oijen, and collaborators developed a single-virion fusion assay four years ago to observe the sequence of events that lead to hemifusion (lipid mixing) and complete fusion (content mixing) induced by influenza virus hemagglutinin (HA) upon a controlled pH drop. In the present work they further developed the assay to also monitor the time it takes for individual virions to stably dock to a supported bilayer (referred to as “arrest”). Furthermore, the assay was streamlined in order to test a number of strategic mutations of HA.

The chosen mutations destabilize the interaction between the fusion peptide and the rest of HA in the pre-fusion state (the structure of which has been determined by X-ray crystallography). The single-virion fusion data suggest that the kinetics of both virion arrest and hemifusion are accelerated by the destabilizing mutations, while the apparent number of events (∼3 trimers) remains unchanged. This result illustrates that release of the fusion peptide from its pocket is rate-limiting for virion arrest and hemifusion, and it is consistent with a model of an extended HA intermediate that bridges membranes prior to a further conformational change that leads to hemifusion.

## Highlight

The key insight in this paper is to make a link between the kinetic intermediates that lead to virion arrest and hemifusion with the structural requirement of fusion peptide release.

## Points to consider in the revisions:

1. A subset of the virons for X31-HA is already arrested prior to the pH drop. Some speculation about the origin of this phenomenon might be warranted. Perhaps some fusion peptides have been released for this variant even at higher pH?

2. What is the efficiency of arrest upon the pH drop for the X31-HA variant? In other words, how many virions transiently dock to the bilayer but do arrest and hemifuse (Movie 1)? Some rough estimation would be sufficient.

3. Likewise, what is the estimated efficiency of hemifusion for those virions that are arrested?

4. As far as we understand it, the final model posits that each trimer releases fusion peptide stochastically, and if 3–4 of them out of about 50 or more trimers do so (regardless of proximity), the virion arrests. The remaining available trimers continue to release fusion peptide at the same individual rate and if this results in peptide release of 3–4 neighboring ones, then the hemi-fusion occurs.

In this model, however, it is not intuitively clear that the kinetics from arrest to hemi-fusion would show a gamma distribution with N ∼ 3–4. The reaction kinetics behind the gamma distribution assumes multiple reaction steps to occur sequentially or parallel. The steps leading to virion arrest may well be described by a gamma distribution because one starts with a large number of trimers and fusion peptide release of the first few would occur with essentially the same global rate. However, the situation is different for the kinetics from arrest to hemi-fusion because the model proposes the release of peptides from 3–4 neighboring trimers.

Could the observed lag-time distributions be reproduced with a simple simulation? For example, one could start with a triangular lattice of a certain size and simulate the entire process based on the proposed model. Each trimer could release peptide with a rate of k, and if N of them release peptide, one calls it an arrest; and then if N of them accumulate next to each other, one calls it hemi-fusion. The simulation could then be repeated with different values for N. Such a simulation would be more directly related to the model that is proposed here.

## Optional

5. It would be nice (but not required) to investigate the effect of the same mutations on content mixing using the content marker that was used in the previous PNAS study. In that previous work, it was concluded that the transition from hemifusion to full fusion is a one step process. One would then expect that the step from hemifusion to full fusion would not be influenced by the HA mutations studied here unless an additional HA trimer is actively engaged to achieve complete fusion.

---

## [Author Response]

*1. A subset of the virons for X31-HA is already arrested prior to the pH drop. Some speculation about the origin of this phenomenon might be warranted. Perhaps some fusion peptides have been released for this variant even at higher pH*?

We have added to the Results and Materials and methods statements to address this observation and speculate about its cause. We have also added two additional movies (Video 9 and Video 10), which speak to this point. In summary, while it is possible and likely that some fusion peptides are released and inserted into the target membrane even at neutral pH (we occasionally observe hemifusion events at neutral pH – see Video 9), we believe that this is a rare occurrence and that most of the pre-arrested events in the context of X31-HA virions at pH 5.5 and above (∼20%) result from imperfections in either the bilayer or the virions themselves.

*2. What is the efficiency of arrest upon the pH drop for the X31-HA variant? In other words, how many virions transiently dock to the bilayer but do arrest and hemifuse (Movie 1)? Some rough estimation would be sufficient*.

*3. Likewise, what is the estimated efficiency of hemifusion for those virions that are arrested*?

Response to points 2 & 3: Virion arrest preceded hemifusion in all cases and hemifusion followed virion arrest in 75% (X31-HA virions) and 65% (UdornS4G virions) of the cases at pH 5.5. We had originally included this measurement in Table 1, and we have now added a sentence in the Materials and methods further discussing these percentages. We also added a sentence about the relative efficiency of arrest, which was nearly 100% up to the highest pH analyzed (pH 5.65). At pH 5.65 we observe a small subset of virions that continue to move under hydrodynamic force well after the pH drop, but in this case also nearly 100% seem to arrest within a few minutes of pH drop. We cannot estimate the exact percentage of those virions that never arrest or that arrest very late, because we do not have a way to automatically detect and track the virions that never arrest, but careful observation of the movies in slow motion allowed us to conclude that ∼100% of virions do arrest at all the pHs we analyzed.

From the subset of virions that were arrested preceding pH drop, 82% (X31-HA virions) and 81% (UdornS4G virions) hemifused at pH 5.5. We have now added these percentages to Table 1.

*4. […] Could the observed lag-time distributions be reproduced with a simple simulation? For example, one could start with a triangular lattice of a certain size and simulate the entire process based on the proposed model. Each trimer could release peptide with a rate of k, and if N of them release peptide, one calls it an arrest; and then if N of them accumulate next to each other, one calls it hemi-fusion. The simulation could then be repeated with different values for N. Such a simulation would be more directly related to the model that is proposed here*.

This was an astute point and an excellent suggestion. We have done the simulation, added a new figure (now Figure 5), and we have added a new section to the Results (titled ‘*Simulation of molecular events in the contact patch between virion and target membrane'*). Briefly, we have generated a computer simulation of our predicted model that virion arrest results from independent triggering and irreversible membrane insertion from several HA trimers *anywhere* at the target membrane interface, and that hemifusion is limited by the same molecular rearrangements, fusion peptide release, and membrane insertion, but requires participation of *neighboring* HA trimers, and is thus related to a different probability. This exercise not only supported our original contention but narrowed down our interpretation of results, allowing us to state that virion arrest results from 3–4 trimers inserting into the target membrane anywhere in the contact patch and hemifusion from 3–4 *neighboring* HA trimers undergoing the same molecular rearrangement. In other words, our observed data are inconsistent with participation of as few as 2 or as many as 5 neighboring HA trimers in hemifusion under our current experimental conditions for a range of relevant contact–patch sizes.

*Optional: 5. It would be nice (but not required) to investigate the effect of the same mutations on content mixing using the content marker that was used in the previous PNAS study. In that previous work, it was concluded that the transition from hemifusion to full fusion is a one step process. One would then expect that the step from hemifusion to full fusion would not be influenced by the HA mutations studied here unless an additional HA trimer is actively engaged to achieve complete fusion*.

This is a good suggestion and one that should be addressed with future work, but we believe it is beyond the scope of the current manuscript.